# Developing a Sustainable Antimicrobial Stewardship (AMS) Programme in Ghana: Replicating the Scottish Triad Model of Information, Education and Quality Improvement

**DOI:** 10.3390/antibiotics9100636

**Published:** 2020-09-23

**Authors:** Jacqueline Sneddon, Daniel Afriyie, Israel Sefah, Alison Cockburn, Frances Kerr, Lucie Byrne-Davis, Elaine Cameron

**Affiliations:** 1Scottish Antimicrobial Prescribing Group, Healthcare Improvement Scotland, Delta House, 50 West Nile Street, Glasgow G1 2NP, UK; 2Pharmacy Department, Ghana Police Hospital, Accra PO Box CT104, Ghana; dspdan77@yahoo.com; 3Department of Pharmacy, Keta Municipal Hospital, Keta-Dzelukope P.O. Box WT82, Ghana; isefah1980@gmail.com; 4NHS Lothian, Western General Hospital, Crewe Road South, Edinburgh EH4-2XU, UK; Alison.Cockburn@nhslothian.scot.nhs.uk; 5NHS Education for Scotland, Glasgow G3 8BW, UK; Frances.Kerr@nes.scot.nhs.uk; 6NHS Lanarkshire, Airdrie ML6 0JS, UK; 7School of Medical Sciences, The University of Manchester, Manchester M13 9PL, UK; lucie.byrne-davis@manchester.ac.uk (L.B.-D.); elaine.cameron@stir.ac.uk (E.C.); 8Division of Psychology, University of Stirling, Stirling FK9 4LA, UK

**Keywords:** antimicrobial stewardship, training, antibiotics use, behavior change

## Abstract

(1) Background: Our aim was to develop robust and reliable systems for antimicrobial stewardship (AMS) in Keta Municipal Hospital and Ghana Police Hospital. Objectives were to build capacity through training staff in each hospital, establish AMS teams, collect data on antibiotic use and support local quality improvement initiatives. (2) Methods: The Scottish team visited Ghana hospitals on three occasions and the Ghanaian partners paid one visit to Scotland. Regular virtual meetings and email communication were used between visits to review progress and agree on actions. (3) Results: Multi-professional AMS teams established and met monthly with formal minutes and action plans; point prevalence surveys (PPS) carried out and data collected informed a training session; 60 staff participated in training delivered by the Scottish team and Ghanaian team cascaded training to over 100 staff; evaluation of training impact demonstrated significant positive change in knowledge of antimicrobial resistance (AMR) and appropriate antibiotic use as well as improved participant attitudes and behaviours towards AMR, their role in AMS, and confidence in using the Ghana Standard Treatment Guidelines and antimicrobial app. (4) Conclusions: Key objectives were achieved and a sustainable model for AMS established in both hospitals.

## 1. Introduction

The Scottish Antimicrobial Prescribing Group (SAPG) has established a comprehensive and robust national antimicrobial stewardship (AMS) programme coordinated by a national group working with regional antimicrobial multi-professional teams [1]. The national group is chaired by an Infection Specialist (Infectious Diseases Consultant or Microbiology Consultant) but the lead for the programme is an Antimicrobial Pharmacist. The regional AMS teams, in common with those in the rest of the UK and other European countries, are generally led by an Infection Specialist but the majority of their stewardship interventions are delivered by Antimicrobial Pharmacists and increasingly supported by specialist nurses. Close multi-professional working has been critical to the success of the Scottish AMS programme. This has been successful in changing prescribing practice, providing rich data on antimicrobial use and resistance, providing education for health and social care staff across all settings and applying quality improvement methodology at scale to tackle areas of poor practice [2]. The approach in Scotland is aligned with and informed by the United Kingdom (UK) Antimicrobial Resistance (AMR) National Action Plan [3] and supports the ambitions for stewardship within Europe [4] and those of the World Health Organisation (WHO) [5] as one of several important actions for tackling AMR.

The model for SAPG was adopted from the Swedish Strategic Programme Against Antibiotic Resistance (Strama) programme [6] following visits by key personnel. The Scottish triad approach utilises Information, Education and Quality Improvement as the three key elements required for effective stewardship. The SAPG model has informed approaches in several other countries including Wales [7], Kenya [8], South Africa [9] and Brazil [10].

In 2019 the SAPG secured a global volunteering grant from the Fleming Fund’s Commonwealth Partnerships for Antimicrobial Stewardship (CwPAMS) [11] to work with two hospitals in Ghana. This was the first such grant that required partnership leads to be pharmacists, reflecting their major role in delivery of AMS. The Ghanaian Ministry of Health had developed national Standard Treatment Guidelines (STG) for the management of common infections and had a 5-year National Action Plan (NAP) for AMR (2017–2021) [12]. The NAP covers improving knowledge of AMR, establishing surveillance of antimicrobial consumption, optimising antimicrobial use, establishment of a functional antimicrobial stewardship (AMS) team in all health facilities in Ghana and supporting sustainable investment in AMR reduction. The implementation of the NAP included, among others, the establishment of a functional AMS team in all health facilities in Ghana, which was lacking [12]. At the time of this study few hospitals in Ghana had progressed with establishing an AMS team or programme. The two hospitals involved in this partnership were keen to progress AMS, management support had been agreed and AMS team members identified.

The SAPG team (antimicrobial pharmacists, antimicrobial nurses, Infectious Disease Consultants and researchers from the University of Strathclyde) created a partnership with lead pharmacists in Ghana Police Hospital (GPH), Accra, and Keta Municipal Hospital (KMH), Volta Region, to support the development of antimicrobial stewardship. These lead pharmacists were supported by medical and nursing managers within their hospitals to provide leadership for a multi-professional AMS team. The project was also supported by health psychologists from The Change Exchange, who provided behavioural science strategies in assessing and changing influences on AMS behaviours [13].

The aim of the project was to develop and implement robust and reliable systems (accountability) and processes (practical tools) for antimicrobial stewardship in GPH and KMH by April 2020. This was to include establishing a local AMS team for each hospital, building capacity through provision of training sessions for a total of up to 25 professionals (medical, pharmacy, nursing and laboratory staff) in each hospital to deliver a local stewardship programme and a supported point prevalence survey (PPS) across each hospital to provide baseline surveillance data on antibiotic use to inform improvements. A simplified behaviour change wheel approach was taken to supplement the SAPG model. In this approach, behaviours are specified, the influences on behaviour are studied and these influences targeted in the intervention [14]. The SAPG triad approach to stewardship (Information, Education and Quality Improvement) was applied with behaviour change concepts incorporated throughout with the aim of developing a robust and crucially sustainable antimicrobial stewardship programme in each hospital.

## 2. Results

### 2.1. Hospital AMS Teams

In advance of the initial visit by the SAPG team the Ghanaian lead pharmacists with their hospital management team each convened a local multi-professional antimicrobial stewardship team to support the project and to ensure long term sustainability in antimicrobial stewardship. A standardised assessment of current stewardship was undertaken in both hospital using a tool developed by the Commonwealth Pharmacy Association (CPA). This identified gaps and informed discussions with the SAPG team during the initial visit. The local AMS teams (specialist doctors, pharmacists and nurses) established regular meetings and acted as champions to promote and engage all professional staff in antimicrobial stewardship. These teams also supported the lead pharmacists on all three elements of the project.

### 2.2. Information

For the initial PPS in May 2019 data were collected from prescription charts and patient notes by the Scotland/Ghana teams from all wards on a single day in each hospital utilising paper-based Global Point Prevalence Survey [15] methodology. Prescriptions were compared for compliance with available STG prior to data entry into the online Global PPS system. The overall prevalence of antibiotic use was 65.0% in GPH and 82.0% in KMH. Prevalence rates ranged from 46.7% to 100.0%, depending on the clinical specialty and patient population (Table 1). Penicillins and other beta-lactam antibiotics were the most prescribed antibiotics in both hospitals, with amoxicillin/clavulanic acid being the most commonly prescribed antibiotic.

Some differences were observed in the quality indicators between the two hospitals (Table 2) however in both hospitals there was good documentation of the indication for antibiotic treatment compared with the benchmark level for African hospitals in the Global PPS. For some indications, guideline compliance could not be assessed especially for antibiotic use for surgical procedures as they were not included in the STG. Where a guideline was available, compliance with the choice of agent was ≥50% in both hospitals for both medical and surgical patients.

No treatment was observed to be based on microbiology data in GPH and were only used for one patient in KMH on the day of survey. Duration of surgical prophylaxis was typically more than one day (GPH 69.0%, KMH 77.0%) with no single dose prophylaxis in either hospital.

Data collection for a follow up PPS was carried out in February 2020 by the Ghanaian teams and results were discussed with the SAPG team. Online data entry and reporting was paused due to the COVID-19 pandemic and will be completed in due course.

### 2.3. Education

#### 2.3.1. Engagement

A total of 60 staff participated in a one day training session held across two days, delivered twice in each hospital by the SAPG team. Nurses made up the majority of participants (22, 36.7%) followed by medical doctors (10, 16.7%) and pharmacists (10, 16.7%). Laboratory scientists, hospital managers, midwives and a public health practitioner made up the remaining 30%.

Feedback forms on the SAPG training were completed by 48 of the 60 participants. Responses were positive with 39 participants rating the session as very good and 9 participants as good.

#### 2.3.2. Knowledge Evaluation

For the knowledge quiz in GPH the participant mean scores were: pre-training 9.2 (SD2.2, range 5–13) and post-training 11.1 (SD1.8) (range 8–13), and in KMH the mean scores were: pre-training 9.4 (SD1.8, range 5–13) and post-training 10.9 (SD1.4) (range 8–13). The mean difference between pre and post-training participant scores in GPH was 1.88 (95% CI 0.753 to 3.008) (*p* = 0.00002) and in KMH the mean difference between the scores was 1.57 (95% CI 0.93 to 2.21) (*p* = 0.00001).

In GPH, training was cascaded by the local AMS team to a total of 25 staff across one session. A total of 18 participants completed the knowledge quiz before a session and 8 participants fully completed it post-training. The mean pre-training score was 8.5/13 (range 6–12) and the mean post-training score was 9.3/13 (range 8–11). During the final visit by the SAPG team, 2 of the original training participants completed a further knowledge quiz (4 months after the training session), scoring 10 and 13 points and total of 8 staff (additional 6 people trained by GPH team) completed the knowledge quiz and scored a mean of 10/13 (range 8–13).

In KMH, training was cascaded by the local AMS team to a total of 144 staff over two training sessions. During the final visit by the SAPG team, 2 of the original training participants completed a further knowledge quiz, scoring 9 and 13 points and a total of 12 staff (who completed SAPG or KMH team training) completed the knowledge quiz and scored a mean of 10.5/13 (range 6–13).

#### 2.3.3. Attitudes and Behaviours Evaluation

Participants from both hospitals demonstrated improved attitudes and behaviours around use of antibiotics after the training session as shown in Table 3 and Table 4. Attitudes and behaviours were similar across professional groups based on comments from the training sessions.

Sustained change in attitudes and behaviours were assessed during the final visit with the following findings: in GPH, staff agreed or strongly agreed with all but two of them having positive stewardship behaviours. Areas where some staff did not agree were the ease of adhering to guidelines and the need for peer support for adherence to guidelines. In KMH, a larger number of staff did not agree with these attitudes towards the guidelines and more staff said they could not access the guidelines.

### 2.4. Quality Improvement

In both hospitals access to guidelines and gaps in local guidance were identified by AMS teams as a key target for improvement. Local guidelines in poster format were developed in collaboration with clinical teams for display in wards and departments to ensure staff were aware of which antibiotics should be used for common infections seen among inpatients. Colour laminated copies of these posters were provided by the SAPG team during the final visit.

In GPH, the AMS team agreed a local action plan with a focus on introducing interns (doctors, pharmacists, nurses) to AMS and developing local guidelines for antibiotic prescribing for wound management, as well as obstetric pre- and post-delivery (Appendix A). Other actions included addressing the need for surveillance and analysis of laboratory antimicrobial data for common infections such as urinary tract infections, initiating routine collection and analysis of antimicrobial prescribing at the outpatient department. All findings were to be shared periodically at clinical meetings and with the drug and therapeutic committee, as well as publishing findings as appropriate.

In KMH the AMS team agreed a local action plan that focused on: rollout of AMS education to all staff; improving the adherence to the local treatment guideline on empirical management of pneumonia for ambulatory patients; and increasing patient awareness (Appendix A). Their long term goal was to create and locally adapt antibiotic policies for KMH. Progress has already been made towards these goals with over 144 staff trained in antimicrobial stewardship locally, patient education initiated in some pharmacy led clinics and an ongoing Quality Improvement project in the out patients department which to date has increased compliance with policy and reduced amoxicillin/clavulanic acid prescribing.

## 3. Discussion

Immense progress has been made with the establishment of a robust and sustainable stewardship in GPH and KMH as a result of this project. Through the expert team from SAPG and The Change Exchange providing practical support and guidance, the Ghanaian lead pharmacists have been able to lead their local AMS teams to gain experience and knowledge of the requirements for a successful AMS programme. Building successful relationships has been key to the success of the project and having a single profession, pharmacists, as leaders has been helpful to demonstrate behaviours and capabilities amongst peers [16]. This will support long-term engagement beyond the project to provide continued advice and guidance as the Ghana AMS programmes mature, as well as potentially supporting the spread of AMS to other hospitals in Ghana. Using a multi-professional approach along with behavior change techniques has also been crucial as stewardship needs to be owned by clinical teams and practiced by all staff to be reliable and sustainable [17].

Regarding the Information element of the project, we demonstrated that the PPS assessment was feasible in both hospitals and can be achieved with limited resources and minimal training of a multi-disciplinary team. Now that staff are familiar with the process, further repeats of PPS will take less time and we are hopeful that eventually direct electronic data collection may be possible to reduce data entry time. The use of repeated PPS is a well-recognised method for measuring both the quantity and quality of antibiotic prescribing where electronic medicine management systems are not available [14]. This will allow progress with improvement work to be tracked and smaller bespoke PPS can also be used to investigate prescribing practice in specific clinical areas or of specific antibiotics.

Our approach to the Education element of the project involved developing training collaboratively to ensure the content met the needs of local clinicians. Delivery of the education by a multi-professional team was successful in imparting knowledge, skills and positive behaviours to support improved use of antibiotics. Key behaviours identified by the psychologists during the first visit around supporting access to guidelines and responsibilities of all staff groups for querying prescriptions that do not follow the guidelines featured in the role play scenarios, giving staff a chance to practice promoted behaviours in a non-threatening way. Participants rated the training highly and the use of lectures and interactive sessions supported good engagement and involvement of everyone in discussion of the issues. The ‘train the trainer’ approach has been successful in building local capacity for provision of ongoing training in both hospitals and potentially beyond to other hospitals in these regions of Ghana. This was demonstrated by the capacity of the local team to train more staff as means of cascading the knowledge of the principles of AMS to untrained staff. The training materials used for the project have been made available via the Commonwealth Pharmacist Association (CPA) website and can be used by others to support similar work. Key learning from the one day sessions was that participants would prefer lectures and interactive sessions to be interspersed rather than have all the lectures at the start. This would also help educators and participants to relax and get to know each other to make the most of the sessions.

The Quality Improvement element of the project was tailored to each hospital’s priorities and ambitions based on the action plans agreed by the AMS teams. Both hospitals identified a need for improved access to guidelines so that staff without a smartphone to access the MicroGuide STG app could easily find the information required when prescribing or administering antibiotics. Locally designed posters proved a useful format and the SAPG team were able to produce a quantity of these for each hospital to support compliance with the guidelines across all wards and departments.

In GPH, the AMS team with obstetrics and gynaecology (OBG) and the surgical unit have developed their antibiotic guidelines for pre- and post-delivery and wound management, respectively. Furthermore, the guidelines for common infections seen at the OBG were developed with guidance from the STG. Currently, routine microbial antibiotic sensitivity data from the laboratory, as well as prescribing of antibiotics audits by the pharmacy department, are being done.

In KMH, weekly prescription analysis of compliance to empirical management of pneumonia of ambulatory patients by pharmacists showed an increasing change in behaviour towards the use of first line antibiotics, and work is ongoing.

Limitations of this project included the limited time spent in Ghana by the SAPG team and by the Ghana team in Scotland. With a large team of 10 experts from SAPG (5 for each hospital) and a limited budget, an intensive schedule was necessary to ensure all three elements of the work were delivered in each hospital. Reliance on email communication and some Skype/WhatsApp calls for discussion of the project was not ideal but in the current climate of virtual meetings and global collaboration that may be the way forward. Time for staff to work as volunteers on the project was also at times difficult to manage as all were full time employees with busy work schedules. A further limitation for the training element is potential bias in data collection as a clear protocol for mandatory participant completion of knowledge and behaviour surveys was not employed.

In the current context of the COVID-19 pandemic, future financial support for exchange visits to support development of AMS in low and middle income countries is unlikely and innovative virtual solutions will be a more feasible approach. There may also be merit in supporting the train the trainer approach employed in this study to spread local expertise for AMS to other Ghana hospitals.

## 4. Materials and Methods

### 4.1. Study Design

The study design was developed in late 2018 and detailed plans were progressed during March and April 2019 following the grant award. Implementation of the three elements of AMS was facilitated by exchange visits during a 9-month period from May 2019 to February 2020. There were three visits by the Scottish team and The Change Exchange to Ghana to support AMS implementation and one visit by the Ghanaian partners to Scotland to observe how AMS has been embedded at local and national level. Regular virtual meetings and email communication were used between visits to review progress, plan training sessions and agree actions. The study did not require ethics approval.

At the initial visit in May 2019 a small multi-professional group from SAPG visited both hospitals and supported data collection on antibiotic use for a baseline PPS using the Global PPS system. At this visit, in both hospitals, the health psychologists interviewed a variety of staff whose behaviours would impact on the use of antimicrobials. This included prescribers and dispensers. These discussions probed the behaviours that would support prescribers and dispensers to improve AMS and the barriers and facilitators to those behaviours.

On the second visit in September 2019, two separate multi-professional groups worked with Ghanaian Partners to provide 2 × 1-day ‘train the trainer’ education in each hospital following a training plan (Appendix A) informed by findings from the initial visit. Interactive training activities were developed using behaviour change principles, including a fun Antibiotic Guardian session where trainees pledged their commitment to AMS actions; an activity identifying barriers to changing practice and problem-solving potential solutions; generating action plans to initiate and maintain changes; role playing potentially difficult conversations with prescribers, patients and families; and practicing using the CwPAMS MicroGuide app to access antimicrobial guidelines. Local pharmacist-led antimicrobial teams agreed an action plan and a Quality Improvement (QI) project.

Local Ghana teams cascaded training to other staff and conducted a second PPS between October 2019 and February 2020.

On the 3rd visit by the SAPG team in February 2020, laminated guideline posters for each hospital were provided to increase access for all staff and progress with local action plans was discussed with the AMS teams to agree next steps. Health psychologists and nurses from the visiting team interviewed a range of healthcare staff at both hospitals to identify changes in AMS behaviours since the trainings and ongoing barriers.

### 4.2. Practical Delivery of the Project

The Global Point Prevalence Survey system [15] was used to collect, submit and generate reports on antibiotic use in each hospital.

Training sessions utilised Microsoft PowerPoint presentations and both plenary and small group workshop discussions. Some elements of the training were filmed using a smartphone camera as a record of AMS pledges made by staff. Staff who attended the training session received a signed certificate of participation. Training was evaluated to assess the change in knowledge and behaviours of participants before and after the session using paper forms. Participants were also asked to complete a paper-based feedback form about their perception of the training session. Participants were not asked for formal consent to use information they provided in the evaluation and feedback forms but consent was presumed from their participation in the training session.

Each facility was encouraged to identify a QI project to address the shortfalls identified in key quality indicators identified by the PPS.

## 5. Conclusions

Key objectives were achieved and a sustainable model for AMS was established in both hospitals. Support for spread of AMS at national level was discussed through partnership meetings with academics in the Medical and Pharmacy Schools, the Ministry of Health AMR lead and Pharmaceutical Society staff with commitment to ongoing collaboration. Overall, members of the SAPG team and the Ghanaian lead pharmacists learned much about each other’s professional practice and countries’ cultures which will remain important memories for all.

## Figures and Tables

**Table 1 antibiotics-09-00636-t001:** Prevalence of antibiotic use in Ghana hospitals compared with Africa data from Global PPS.

Prevalence of Antibiotic Use	Adult Total	Paediatric Total
Ghana Police Hospital % (*n* = 59)	57.1 (49)	76.9 (10)
Keta Municipal Hospital % (*n* = 101)	55.6 (90)	100.0 (11)
Africa (Global PPS) % (70 hospitals)	64.2	79.4

**Table 2 antibiotics-09-00636-t002:** Quality indicators for antibiotic use in Ghana hospitals compared with Africa data from Global PPS.

Quality Indicator	Ghana Police Hospital % (*n* = 59)		Keta Municipal Hospital % (*n* = 101)		Africa Global PPS in 70 Hospitals	
	Medical	Surgical	Medical	Surgical	Medical	Surgical
Indication for antibiotic use recorded	100	85	88	84.5	60.8	57.6
	(41)	(17)	(66)	(11)	(1839)	(1230)
Guidelines missing	46.3	70	1.3	46.2	24.1	43.9
	(19)	(14)	(1)	(6)	(729)	(938)
Guideline compliant	62.5	66.7	55.4	50	55.9	61.2
	(10)	(4)	(31)	(2)	(670)	(370)
Stop/review date in notes	92.7	95	98.7	100	29.1	32.4
	(38)	(19)	(74)	(13)	(880)	(693)

**Table 3 antibiotics-09-00636-t003:** Pre and post education responses to survey questions by staff at GPH (Ghana Police Hospital).

Statement		Strongly Disagree	Disagree	Neither Agree nor Disagree	Agree	Strongly Agree	Don’t Know
Antimicrobial resistance (AMR) is a serious problem	Pre				4	15	
	Post	1			4	14	
I am worried that antibiotics will soon become ineffective	Pre	1	1	1	7	8	1
	Post	1			4	13	1
I am worried patients will develop antibiotic resistant infections	Pre	1		1	7	10	
	Post	2			6	11	
Following national or local antibiotic prescribing guidelines will help to prevent the development of AMR	Pre			2	8	9	
	Post	1			5	13	
It is part of my professional role to reduce the risks of AMR	Pre	1			5	13	
	Post				5	13	1
I am able to access the GSTG easily	Pre	1			6	12	
	Post			6	6	6	1
I find it easy to adhere to GSTG whenever I prescribe or administer antimicrobials	Pre	1			9	9	
	Post		1	5	8	4	1
My peers support adherence to GSTG when prescribing or administering antimicrobials	Pre	1		2	10	5	1
	Post			7	11		
I feel confident about questioning a colleague about an antibiotic prescription not in line with the GSTG	Pre	1		4	11	3	
	Post		2	4	11	1	1
I plan to adhere to GSTG whenever I prescribe or administer an antibiotic	Pre	1			10	7	1
	Post			1	10	7	1

GSTG—Ghana Standard Treatment Guidelines.

**Table 4 antibiotics-09-00636-t004:** Pre and post education responses to survey questions by staff at KMH (Keta Municipal Hospital).

Statement		Strongly Disagree	Disagree	Neither Agree nor Disagree	Agree	Strongly Agree	Don’t Know
Antimicrobial resistance (AMR) is a serious problem	Pre	2			4	21	1
	Post					28	
I am worried that antibiotics will soon become ineffective	Pre	1	1		11	14	1
	Post		1		1	26	
I am worried patients will develop antibiotic resistant infections	Pre	2	1		13	12	
	Post				4	23	1
Following national or local antibiotic prescribing guidelines will help to prevent the development of AMR	Pre	2	1	1	10	14	
	Post				3	25	
It is part of my professional role to reduce the risks of AMR	Pre	2			10	15	1
	Post				1	27	
I am able to access the GSTG easily	Pre	3	6	4	12	2	1
	Post		3	1	10	14	
I find it easy to adhere to GSTG whenever I prescribe or administer antimicrobials	Pre	1	2	10	14	1	
	Post		1	4	7	16	
My peers support adherence to GSTG when prescribing or administering antimicrobials	Pre		6	14	6	2	
	Post		2	5	13	8	
I feel confident about questioning a colleague about an antibiotic prescription not in line with the GSTG	Pre	2	4	11	9	2	
	Post	1		1	9	17	
I plan to adhere to GSTG whenever I prescribe or administer an antibiotic	Pre	2		3	16	7	
	Post			1	2	25	

GSTG—Ghana Standard Treatment Guidelines.

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
