# Peer review of "Developing a Sustainable Antimicrobial Stewardship (AMS) Programme in Ghana: Replicating the Scottish Triad Model of Information, Education and Quality Improvement"

_antibiotics, 2020, doi:10.3390/antibiotics9100636_

Round 1
Reviewer 1 Report
v Strengths The topic of this article has quite scientific soundness, and the research here conducted gives solid evidences. Still, methods have to be better described. In addition, further minor ethical concerns should be better detailed. English is remarkably fine.
|
v Weaknesses Title and keywords: Consider if “training” could be more appropriate than “education” in this specific context. Introduction: Several approaches are not less important than antimicrobial stewardship to address antimicrobial resistance. Even though this paper is specific for AMS, please consider expanding this section. You may want, for instance, to consider these articles: DOI: 10.3390/ijerph13100940. Messina G et al. “Time effectiveness of Ultraviolet C light (UVC) emitted by Light Emitting Diodes (LEDs) in reducing stethoscope contamination” (IJERPH 2016) DOI: 10.1016/j.nedt.2017.05.011. Lawson C et al. “Development of an international comorbidity education framework” (NEDT 2017)
Results: Line 97 and 183: Consider writing, for the best readability, instead of the more colloquial “co-amoxiclav”, the full name “amoxicillin/clavulanic acid”. Line 105: Consider providing any benchmark in order to make this statement better understandable and comparable. Tables: Consider thickening some borders, in order to enhance readability, for example between different “statements” (tables 3-4) Table 3: Consider not splitting between pages two rows that the reader would most likely compare (in this case a “pre” row is in page 4 and a “post” row is in page 5). Discussion: Please, identify whether there are some possible collection BIAS due the fact that a clear protocol is not declared. Materials and Methods: Like you have well described the intervention steps and their timing, please specify as well the design of the study and when it was conducted, since it is not clear. Please declare whether this intervention did not require ethics committee approval, or, in case it underwent evaluation, provide reference of its approval. Were participants asked for consent to be included in this study? If so, please declare it. |
Author Response
Strengths
The topic of this article has quite scientific soundness, and the research here conducted gives solid evidences. Still, methods have to be better described. In addition, further minor ethical concerns should be better detailed.
English is remarkably fine.
Thanks you for your comments on the strengths of the study.
Weaknesses
Title and keywords:
Consider if “training” could be more appropriate than “education” in this specific context.
We agree that training is a more appropriate term for the intervention and have changed this in the keywords. (line 36)
Introduction:
Several approaches are not less important than antimicrobial stewardship to address antimicrobial resistance. Even though this paper is specific for AMS, please consider expanding this section. You may want, for instance, to consider these articles:
DOI: 10.3390/ijerph13100940. Messina G et al. “Time effectiveness of Ultraviolet C light (UVC) emitted by Light Emitting Diodes (LEDs) in reducing stethoscope contamination” (IJERPH 2016)
DOI: 10.1016/j.nedt.2017.05.011. Lawson C et al. “Development of an international comorbidity education framework” (NEDT 2017)
Thank you for your comment. We acknowledge that AMS is not the only approach to tackling AMR and have added that other actions are also important. (lines 44-46) We have not added additional references as other actions are detailed within the WHO reference already cited.
Results:
Line 97 and 183: Consider writing, for the best readability, instead of the more colloquial “co-amoxiclav”, the full name “amoxicillin/clavulanic acid”.
We have amended as suggested. (lines 99 and 185)
Line 105: Consider providing any benchmark in order to make this statement better understandable and comparable.
Thank you for your comment. We have reflected on the Ghana hospitals’ performance compared with the African benchmark. (line 107)
Tables: Consider thickening some borders, in order to enhance readability, for example between different “statements” (tables 3-4)
We have amended as suggested.
Table 3: Consider not splitting between pages two rows that the reader would most likely compare (in this case a “pre” row is in page 4 and a “post” row is in page 5).
We have amended as suggested.
Discussion:
Please, identify whether there are some possible collection BIAS due the fact that a clear protocol is not declared.
Thank you for your comment. We have added a sentence to acknowledge this potential bias. (lines 247-249)
Materials and Methods:
Like you have well described the intervention steps and their timing, please specify as well the design of the study and when it was conducted, since it is not clear.
Thank you for your comment. We have added a sentence to provide further details of the study design and dates when it was conducted. (lines 255-261)
Please declare whether this intervention did not require ethics committee approval, or, in case it underwent evaluation, provide reference of its approval.
We have added a sentence to clarify that this work did not require ethics committee approval. (line 263)
Were participants asked for consent to be included in this study? If so, please declare it.
Thank you for your comment. We have added a sentence regarding presumed consent via participation in the training session. (lines 294-296)
Reviewer 2 Report
General comment :
It is a research / action investigation describing the process of training (or implementation? See Alinea 1) of antimicrobial stewardship (AMS) in two Ghana hospitals based on the experience of AMS in aother country (Scotland).
Some information is lacking:
- Was there any form of AMS in these two Ghanean hospitals before this cooperation program? It should be clearly stated whether it is a new process (AMS), whether it is an improvement of AMS already existing AMS (even with another name) – and in this last case, which was already done in PPS, for example, and the new improvements associated with the program ; the dynamic of new organisational items should be better described
- Some major differences between AMS management / organisation in Scotland and in Ghana should be openly described: AMS mainly under leadership of MD’s with infectious disease specialty in Scotland and under pharmacist leadership in Ghana? If yes, is there an obvious reason for this difference? Maybe in Ghana are MD’s not full-time hospitalists, at the difference of pharmacists (I don’t know), and this can explain the difference. From European countries point of view, MD’s specialists in infectious disease are frequently leaders of AMS process as they are the specialists prescribing antibiotics. Of course, it a a sensitive point, and cooperation between pharmacists and MD’s is everywhere as essential as difficult… But a few management words on this aspect should clarify the mode of management of AMS and should be openly described with its justification;
- Is there no impact on financial support requirement for such an improvement of AMS? You evoke the lack of financial support for more exchanges between Scottish and Ghanaian hospitals, but you don’t say anything about the requirements of more qualified personnel for AMS in Ghana hospitals.
COVID-19 has shown that many interactions can be adequately developed on line – limiting the financial cost. Such a cooperation program should be clearly maintained by taking advantage of new mode of Ghana/Scotland interaction. The passage where you ask more financial support for formal exchanges of pople between Scotland and Ghana should be rewritten with a new COVID-pandemic way of changing our habits. Personnaly, I felt a little shocked by this passage in the context of COVID-19 pandamic.
Author Response
General comment :
It is a research / action investigation describing the process of training (or implementation? See Alinea 1) of antimicrobial stewardship (AMS) in two Ghana hospitals based on the experience of AMS in aother country (Scotland).
Thank you for your comment. We intended the manuscript to describe the process of implementing stewardship in Ghana following our Scottish model. I hope this clarifies your query.
Some information is lacking:
Was there any form of AMS in these two Ghanean hospitals before this cooperation program? It should be clearly stated whether it is a new process (AMS), whether it is an improvement of AMS already existing AMS (even with another name) – and in this last case, which was already done in PPS, for example, and the new improvements associated with the program ; the dynamic of new organisational items should be better described
Thank you for your comment. We have expanded the section about status of AMS in the hospitals prior to this partnership project. (lines 60-63)
Some major differences between AMS management / organisation in Scotland and in Ghana should be openly described: AMS mainly under leadership of MD’s with infectious disease specialty in Scotland and under pharmacist leadership in Ghana? If yes, is there an obvious reason for this difference? Maybe in Ghana are MD’s not full-time hospitalists, at the difference of pharmacists (I don’t know), and this can explain the difference. From European countries point of view, MD’s specialists in infectious disease are frequently leaders of AMS process as they are the specialists prescribing antibiotics. Of course, it a a sensitive point, and cooperation between pharmacists and MD’s is everywhere as essential as difficult… But a few management words on this aspect should clarify the mode of management of AMS and should be openly described with its justification;
Thank you for your comment. We have expanded the section about leadership for AMS in Scotland and Ghana and also the reasons for pharmacist leadership in this project. (lines 40-46, 59-61, 74-76)
Is there no impact on financial support requirement for such an improvement of AMS? You evoke the lack of financial support for more exchanges between Scottish and Ghanaian hospitals, but you don’t say anything about the requirements of more qualified personnel for AMS in Ghana hospitals.
Thank you for your comment. We have added a sentence about the challenge of funding AMS and the ambition for all clinical practitioners to engage with AMS through additional training. (lines 264-267)
COVID-19 has shown that many interactions can be adequately developed on line – limiting the financial cost. Such a cooperation program should be clearly maintained by taking advantage of new mode of Ghana/Scotland interaction. The passage where you ask more financial support for formal exchanges of pople between Scotland and Ghana should be rewritten with a new COVID-pandemic way of changing our habits. Personnaly, I felt a little shocked by this passage in the context of COVID-19 pandamic.
Thank you for your comment. I am unable to find a passage about asking for more financial support. However I have emphasised in the Discussion (lines 264-267) that virtual communication and innovative solutions which have become common due to the covid-19 pandemic have potential to support further AMS initiatives utilising our model.
Round 2
Reviewer 1 Report
The manuscript has been significantly improved and I think now there aren't any flaws.